# *Roseburia intestinalis* and Its Metabolite Butyrate Inhibit Colitis and Upregulate TLR5 through the SP3 Signaling Pathway

**DOI:** 10.3390/nu14153041

**Published:** 2022-07-25

**Authors:** Guangcong Ruan, Minjia Chen, Lu Chen, Fenghua Xu, Zhifeng Xiao, Ailin Yi, Yuting Tian, Yi Ping, Linling Lv, Yi Cheng, Yanling Wei

**Affiliations:** 1Department of Gastroenterology, Daping Hospital, Army Medical University (Third Military Medical University), Chongqing 400042, China; ruanguangcong@163.com (G.R.); chenminjia525@163.com (M.C.); chenlu130652@163.com (L.C.); xufenghua0603@126.com (F.X.); xiaozf622@126.com (Z.X.); yalyal120@163.com (A.Y.); 15002372386@163.com (Y.T.); pingyi2022@126.com (Y.P.); lvling0827@163.com (L.L.); 2Department of Pathogenic Biology and Immunology, School of Basic Medicine, Ningxia Medical University, Yinchuan 750004, China

**Keywords:** *Roseburia intestinalis*, butyrate, TLR5, Sp3, ulcerative colitis

## Abstract

The pathogenesis of ulcerative colitis (UC) is unclear, but it is generally believed to be closely related to an imbalance in gut microbiota. *Roseburia intestinalis* (*R. intestinalis*) might play a key role in suppressing intestinal inflammation, but the mechanism of its anti-inflammatory effect is unknown. In this study, we investigated the role of *R. intestinalis* and Toll-like receptor 5 (TLR5) in relieving mouse colitis. We found that *R. intestinalis* significantly upregulated the transcription of TLR5 in intestinal epithelial cells (IECs) and improved colonic inflammation in a colitis mouse model. The flagellin of *R. intestinalis* activated the release of anti-inflammatory factors (IL-10, TGF-β) and reduced inflammation in IECs. Furthermore, butyrate, the main metabolic product secreted by *R. intestinalis*, regulated the expression of TLR5 in IECs. Our data show that butyrate increased the binding of the transcription factor Sp3 (specificity protein 3) to the TLR5 promoter regions, upregulating TLR5 transcription. This work provides new insight into the anti-inflammatory effects of *R. intestinalis* in colitis and a potential target for UC prevention and treatment.

## 1. Introduction

Ulcerative colitis (UC) is a chronic inflammatory bowel disease that is characterized by symptoms such as abdominal pain, diarrhea, and weight loss. In recent decades, it has been found that the pathogenesis of UC may be related to genetics, gut microbiota, immunodeficiency, or environmental factors [1,2]. Due to the complexity of pathogenesis and incurable characteristics, it is of great importance to further explore pathogenesis and to discover new treatment methods for the clinical treatment of UC.

Increasing evidence indicates that gut microbes are vital to the host, and changes in intestinal homeostasis may lead to UC. Numerous studies have suggested that the gut microbiota plays a central role in UC etiology [3,4]. Germ-free mice displayed UC-like symptoms after receiving fecal microbiota from UC patients [5]. As one of many new treatments, some studies have reported that microbiota based treatments have significant therapeutic effects on UC by probiotics or fecal microbiota transplantation (FMT), although the effective microorganisms (EM) and the mechanisms for this therapeutic effect are poorly understood [3,6,7]. In mice with dextran sulfate sodium (DSS)-induced chronic colitis, gut inflammation and dysbiosis are ameliorated with a high dosage of *Bacillus subtilis* [8]. Studies have shown that *Bifidobacterium longum 536* improves clinical signs in UC patients with mild-to-moderate disease [9]. One study suggested that *R. intestinalis* induced an anti-inflammatory response and released anti-inflammatory factors to control Crohn’s disease (CD) [10]. Another project demonstrated that *Roseburia spp.* modulated the intestinal microbiota, reduced the occurrence of leaky gut, and further improved the manifestation of alcohol-related liver diseases [11]. Moreover, some studies have reported that short-chain fatty acids derived from intestinal microbes are involved in maintaining intestinal homeostasis [12,13]. Short-chain fatty acids (SCFAs) also play an important role in the resolution of inflammation [14,15,16,17]. In our previous studies, we verified that *R. intestinalis* also has therapeutic potential in the treatment of UC. The administration of *R. intestinalis* in mice with DSS-induced colitis contributed to the restoration of the gut microbiota, promoted colon repair, and reduced the severity of colitis [18]. We wondered how these bacteria alleviate the inflammation of UC and the underlying molecular mechanism.

The administration of *R. intestinalis* can effectively change the structure of the host intestinal microbiota and their metabolites to coordinate the intestinal immune microenvironment, particularly with regard to butyrate [11,19,20]. Butyrate plays a crucial role in maintaining colonic homeostasis and has multiple effects on mammalian cells. These actions include inhibiting proliferation, inducing differentiation, and regulating gene expression [21,22]. Histones in cells acetylated with a butyrate treatment revealed that butyrate inhibits histone deacetylase (HDAC) activity [23]. Recombinant Toll-like receptor 5 (TLR5), the only receptor of flagellin, is widely expressed in intestinal epithelial cells and plays an important role in the inflammatory response. Compared with normal tissues, the expression of TLR5 was significantly downregulated in UC lesions [24]. Butyrate and flagellin are both involved in the regulation of inflammation, and whether they have a synergistic effect through TLR5 remains to be determined. Thakur et al. discovered that butyrate can initiate TLR5 transcription by regulating specificity protein 1, 3 (SP1/SP3) [25]. What is unknown is whether there are more signaling pathways that regulate the expression of TLR5 by butyrate. The expression of butyrate-responsive genes with butyrate response elements in the promoters can often be mediated by butyrate through Sp1/Sp3 transcription factor binding sites [26]. NHE8 (sodium/hydrogen exchanger, NHE), the newest member of the intestinal NHE family, has been suggested to be involved in sodium absorption in the intestinal tract. Butyrate participates in the regulation of NHE8 expression and enhances the absorption of sodium. In this process, the Sp3 transcription factor can be induced by butyrate to bind with the NHE8 promoter to activate the transcription of NHE8 [27]. One study demonstrated that butyrate, as a histone deacetylase inhibitor, can be used to upregulate the transcription of p21/WAF1/CIP1 through Sp3 but not through Sp1 [28]. The expression of TLR5 is significantly downregulated in UC lesions compared with normal tissues [24]. Further study is required to determine whether butyrate contributes to the increased expression of TLR5 or Sp1/Sp3.

Herein, the efficacy of *R. intestinalis* in ameliorating DSS-induced inflammation in mice was reported. Notably, exogenous treatment with *R. intestinalis* flagellin significantly activated the TLR5-mediated anti-inflammatory pathway. In addition, as an important metabolite of *R. intestinalis*, butyrate can further upregulate the expression of TLR5 by inducing the binding of the transcription factor Sp3 to the TLR5 promoter. These results provide significant insights into the role of *R. intestinalis* in UC disease resistance and inflammatory disease control in the intestine.

## 2. Materials and Methods

### 2.1. Preparation of R. intestinalis

*R. intestinalis* (DSMZ-14610) (Deutsche Sammlung von Mikroorganismen und Zellkulturen GmbH, Braunschweig, Germany) was used in this experiment and frozen in cryotubes at −80 °C. The bacteria were grown in BD BACTEC Lytic/10 Anaerobic/F medium (Becton Dickinson and Company, Franklin Lakes, NJ, USA) for 24 h at 37 °C under strictly anaerobic conditions when the solid phase was achieved.

### 2.2. Animals and Groups

Male C57BL/6 mice were purchased from Vital River Laboratories (Beijing, China). Mice aged 6 to 8 weeks were used in all experiments. The mice were raised and kept in specified pathogen-free (SPF) conditions at the Experimental Animal Center, Institute of Field Surgery, Committee of Army Medical Center. The mice were housed at a temperature of 22 °C with a 12:12 h light–dark cycle and had free access to food and water. All animal-related procedures followed guidelines approved by the Laboratory Animal Welfare and Ethics Committee of Third Military Medical University (AMUWEC2020639).

The mice were randomly assigned to one of four groups (*n* = 6): control, DSS, DSS + *R. I*, or DSS + butyrate. Mice in the control group were reared under standard conditions. Butyrate (B5887) was purchased from Sigma (St. Louis, MO, USA). Colitis was induced in the mice of the DSS group by administering 2% (*w*/*v*) DSS for 7 d. Mice in the DSS + *R. I*. group were treated with 2% dextran sulfate sodium salt (DSS, MP Biomedicals, Irvine, CA, USA) for 7 d and then treated with 200 μL of 1 × 10^9^ CFU *R. intestinalis* [18]. Mice in the DSS + butyrate group were treated with 2% dextran sulfate sodium salt for 7 d and then treated with 100 μg/g butyrate [22]. Competent authorities reviewed and authorized all animal experiments. All mice were euthanized, and colon tissues were promptly collected. The colon tissues were used for Western blotting or RNA extraction. The morphologies of samples of the distal colon that were fixed, embedded, and stained with hematoxylin-eosin (H&E) were examined under a microscope.

### 2.3. Disease Activity Index

The disease activity index (DAI) scores for mice were calculated as the sum of three factors: weight loss, ranging from 0 to 4 (0, no loss; 1, 1 to 5% loss; 2, 6 to 10% loss; 3, 11 to 15% loss; and 4 for weight loss greater than 16%); stool traits, 0 to 4 (0, normal; 2, loose stools; and 4, diarrhea); and stool bleeding, ranging from 0 to 4 (0, no blood; 2, stool occult blood positive; and 4, grossly bloody stool) [29].

### 2.4. Extraction of Flagellin

Flagellin was extracted from bacteria using a previously described method. Briefly, bacteria were collected by centrifugation and resuspended in prechilled PBS. The cell suspension was then homogenized 3 times in a blender for 30 s each time at high speed. Cell fragments were removed by centrifugation (10,000× *g* for 20 min at 4 °C), and crude flagellin was concentrated by ultracentrifugation (100,000× *g* for 1 h at 4 °C). The resultant pellets were resuspended in 100–200 μL sterile distilled water, and the protein content was determined using the Pierce BCA protein assay (Thermo Fisher, Waltham, MA, USA) following the manufacturer’s instructions.

### 2.5. Cell Culture

DMEM (Gibco, Thermo Fisher, Waltham, MA, USA) supplemented with 10% fetal bovine serum (Gibco, Thermo Fisher, Waltham, USA) containing 100 IU/mL penicillin and 100 μg/mL streptomycin was used to culture HT29 cells, which were purchased from the American Type Culture Collection. Cells were incubated at 37 °C in a humidified incubator with 5% CO_2_. Cell-stimulating drugs such as butyrate and lipopolysaccharide (LPS) as well as actinomycin D (ActD) and cycloheximide (CHX) were purchased from Sigma (St. Louis, MO, USA).

### 2.6. RT–qPCR and Western Blotting

Total RNA was extracted from the colon and bacterial cells with TRIzol reagent (Takara, Kusatsu, Shiga, Japan) following the manufacturer’s protocol. First-strand cDNA was synthesized from 1 μg of total RNA using the QuantiNova Reverse Transcription Kit (Takara, Kusatsu, Shiga, Japan). Reverse transcription-quantitative PCR (RT–qPCR) was performed using SYBR^®^ Green I (Takara, Kusatsu, Shiga, Japan). A melting curve analysis was used to determine the specificity of the PCR products, and the 2-ΔΔCt method was used to determine the relative expression of the target gene (Table 1).

Proteins from intestinal tissues and bacterial cells were extracted with RIPA lysis buffer (KeyGen Biotech, Nanjing, China) containing phosphatase and protease inhibitors, and the concentration was measured with a BCA protein quantification kit (KeyGen Biotech, Nanjing, China). The blocked membranes were incubated with specific primary antibodies (GAPDH, 1:2000, ab181602, Abcam, Cambridge, MA, USA; β-actin, 1:2000, 3700S, CST, Danvers, MA, USA; TLR5, 1:1000, ab13876, Abcam, Cambridge, MA, USA; and Sp3, 1:1000, sc-28305, Santa Cruz, Dallas, TX, USA) overnight at 4 °C. Corresponding HRP-labeled IgG (IgG; 1:1000, ab172730 Abcam, Cambridge, MA, USA) was applied to detect the primary antibody on PVDF membranes. The sections were developed with a detection kit (Thermo Fisher, Waltham, MA, USA).

### 2.7. IHC Staining

The subcellular localization of proteins was measured by immunohistochemistry (IHC staining). Antibodies (MPO, 1:200, AF3667, R&D, Minneapolis, MN, USA; Sp3, 1:200, A12790, Abclonal, Wuhan, China; and TLR5, 1:200, ab13876, Abcam, Cambridge, MA, USA) were applied to detect the location of target proteins. In short, colon specimens were fixed with 4% paraformaldehyde and embedded in paraffin. All the specimens were serially sectioned followed by high-temperature and high-pressure repair. The paraffin sections were incubated with antibodies, and then DAB staining was performed.

### 2.8. Immunofluorescence

Immunofluorescence was applied to detect the level of TLR5 protein. Cell slides were fixed with paraformaldehyde, rinsed in PBS, and blocked with goat serum at room temperature. Blocked cell slides were incubated with specific antibodies in wet boxes overnight at 4 °C. Slides were washed with PBST and then incubated with antifluorescence at room temperature for 1 h. DAPI was added dropwise to stain the specimens, and fluorescence signals were observed by fluorescence microscopy.

### 2.9. ELISA

IL-10 (Cat. number PI528) and TGF-β (Cat. number PT880) proteins were detected by ELISA kits according to the procedures in the experimental guidelines (Beyotime, Nantong, China). The minimum detection amounts were 8.8 pg/mL and 1.8 pg/mL, respectively.

### 2.10. Cytokine Quantitation

At 14 d following induction, the colon was harvested, and the weight was recorded. Tissue samples were lysed in NP-40 lysis buffer (5 μL/mg) containing a protease inhibitor cocktail (Beyotime, Nantong, China). The lysate was separated and stored at −80 °C until further use. A Multifactor Analysis Kit (Biolegend, San Diego, CA, USA) was used to analyze the samples in accordance with the manufacturer’s instructions.

### 2.11. siRNA Interference

Sp3 siRNA and control siRNA, purchased from Santa Cruz Biotechnology, were transfected into HT29 cells with riboFECT mRNA Transfection Reagent (RiboBio, Wuhan, China). The silencing efficiency was measured 48 h after transfection by RT–qPCR and Western blotting.

### 2.12. ChIP–qPCR

Following the manufacturer’s instructions, chromatin immunoprecipitation (ChIP) was carried out using an EZChIPTM chromatin immunoprecipitation kit (CST, Danvers, MA, USA). In brief, HT29 cells were collected and cross-linked using formaldehyde following various treatments. After digestion with a nuclease, the genomic DNA was digested. The chromatin was immunoprecipitated with Sp3 and IgG antibodies overnight and then incubated with Protein G agarose for one hour. The complexes were then removed from the elution buffer and washed sequentially with low salt, high salt, LiCl, and Tris-EDTA buffer. The DNA–protein complexes were reversed, and ethanol was used to purify the DNA. qPCR was used to measure the binding of the TLR5 promoter to proteins.

### 2.13. Analysis of Fecal SCFAs

Concentrations of fecal SCFAs were determined by gas chromatography/mass spectrometry (GC/MS). Briefly, 600 mg of feces was homogenized with 1.2 mL of phosphate buffer (pH 7.3) and centrifuged at 4 °C at 16,000× *g* for 15 min. The supernatants were filtered through a 0.22 µm nylon filter (EMD Millipore, St. Louis, MO, USA). An aliquot (200 µL) of the supernatant was acidified by adding 0.1 mL of 50% (*v*/*v*) sulfuric acid. After vortexing and standing for 2 min, the organic acids were extracted by adding 0.4 mL of diethyl ether, and the supernatants were measured by GC/MS on a 7 Agilent 6890 (Agilent Technologies, Santa Clara, CA, USA) equipped with flame ionization, thermal conductivity detectors, capillary columns, and GC Chem Station software (version E.02.00.493, Agilent Technologies, JSB, Eindhoven, The Netherlands).

### 2.14. Statistical Analyses

All statistical analyses were repeated three times independently, and all results were reproducible. GraphPad Prism 8 (GraphPad Software Inc., San Diego, CA, USA) was used to conduct the statistical analyses. To make statistical comparisons, a t-test or one-way analysis of variance (ANOVA) was used. *p* < 0.05 was considered statistically significant.

## 3. Results

### 3.1. DSS-Induced Intestinal Inflammation Relieved by R. intestinalis

*R. intestinalis* was cultured and identified before treatment. Figure A1A shows the results of the strain identification. Scanning electron microscopy of *R. intestinalis* is shown in Figure A1B. The efficacy of *R. intestinalis* therapy was then investigated in a mouse model of colitis in which colonic inflammation was induced using DSS (Figure 1A). In comparison with the control group, the mouse body weights in the DSS and DSS + *R. I* groups fed 2% DSS were considerably decreased in the first 7 d (Figure 1B). However, the mouse body weights in both the DSS group and DSS + *R. I* groups increased gradually without the 2% DSS solution, particularly in the DSS + *R. I* (Figure 1B). Moreover, treatment with *R. intestinalis* significantly reduced the DAI score (Figure 1C) and recovered the colon length (Figure 1D,F) in the DSS + *R. I* group.

Damage to the mucosal barrier is the initial event in chemically induced colitis models, and the colonic mucosa morphology was evaluated by H&E staining (Figure 1E,G). Sparse short villi and a thin intestinal mucosa were observed in DSS-treated mouse colons. Additionally, mucosal and submucosal inflammatory cells, swollen crypts and crypt destruction, and damaged epithelial cells were also observed in the DSS group. These pathological features were improved significantly in mice after receiving the *R. intestinalis* treatment. The anti-inflammatory effect of *R. intestinalis* on colitis was also verified by MPO staining of colon tissue (Figure 1E,H). These findings showed the curative effect of *R. intestinalis* administration on DSS-induced colitis in mice, and *R. intestinalis* may have therapeutic potential for UC.

### 3.2. Flagellin from R. intestinalis Displayed Anti-Inflammatory Effects in HT29 Cells

To study the anti-inflammatory effect of *R. intestinalis* on the intestinal mucosa, we used LPS, an endotoxin in the cell wall of Gram-negative bacteria, on IECs as an in vitro model of the inflamed colon mucosa. To investigate regulation by *R. intestinalis* on inflammatory IECs, inflammatory IECs induced by LPS (1 μg/mL) stimulation were treated in vitro with *R. intestinalis* (1 × 10^8^ CFU/1 × 10^6^ HT29 cells) and *R. intestinalis* flagellin (1 μg/mL) (Figure A1C) for 24 h (Figure 2A). The TLR5 receptors in IECs could be activated by *R. intestinalis* flagellin, which then acts as an anti-inflammatory signal. After treatment with *R. intestinalis* and flagellin, the secretions of the anti-inflammatory factors IL-10 and TGF-β by IECs were significantly increased (Figure 2B–E). Compared with the untreated group, the mRNA expression and protein levels of IL-10 and TGF-β were significantly decreased in cells cocultured with LPS for 24 h. However, the expression and protein levels of the inflammatory factors in LPS-treated cells were restored to normal levels in cells supplemented with *R. intestinalis* or the extracted flagellin. These results suggest that *R. intestinalis* as well as its flagellin can induce the expression of anti-inflammatory factors against inflammation.

### 3.3. Determination of TLR5 Expression and Screening of Related Transcription Factors

TLR5 plays a crucial role in the immune response. An immune response can be induced once the flagellin of pathogenic bacteria is recognized by TLR5. *R. intestinalis* and its flagellin have been proven to contribute to inhibiting UC in mice by triggering the expression of the anti-inflammatory factors IL-10 and TGF-β. Thus, the expression levels of TLR5 and the related transcription factor Sp3 in the epithelial cells of mice treated with *R. intestinalis* were investigated. Compared with the other two groups, TLR5 and the potential transcription factor Sp3 were reduced in the epithelial cells of mice with DSS-induced colitis (Figure 3A). The decrease in TLR5 levels was consistent with the reduction in Sp3 expression levels (Figure 3B), which was also confirmed by IHC staining for TLR5 and Sp3 in the intestinal mucosa (Figure 3D). There was no significant difference in Sp1. Total proteins were extracted from colon tissue, and the levels of TLR5 and Sp3 were detected by Western blot (Figure 3E); these values were consistent with the qPCR results. These results indicated that TLR5 levels were significantly decreased in DSS-induced UC mice. The increased expression of TLR5 in UC mice and the remission of UC were mediated by *R. intestinalis*, which also suggested a potentially important role of TLR5 in UC amelioration.

### 3.4. R. intestinalis Inhibited the Inflammatory Progression of UC through Its Metabolite Butyrate

To explore the role of *R. intestinalis* in the treatment of UC, particularly the anti-inflammatory effect, the feces of all treated mice were collected, and metabolomic sequencing was performed. The metabolism of short-chain fatty acids decreased under colitis conditions but increased after treatment, particularly in butyric acid-treated mice (Figure 4A). These results indicate that butyric acid may play an important role in the anti-inflammatory effect of *R. intestinalis* on UC; therefore, both *R. intestinalis* and butyrate were used to treat the colitis mice in further studies (Figure 4B).

Compared with the control group, the weight of the colitis mice decreased significantly in the first 7 d, and the butyrate treatment improved the weight loss (Figure 4C) and the DAI (Figure 4D) score, which were similar to the *R. intestinalis* treatment group. At the end of the observation period, the colon lengths of the butyrate treatment group and the *R. intestinalis* treatment groups were 1.2 and 1.11 times those of the DSS group, respectively (Figure 4E). We also quantified the expression of inflammatory factors in colon tissues to identify the anti-inflammatory effect of the two treatments on colitis. The cytokine levels (IL-6, IFN-γ, and TNF-α) in the colitis mouse colon on Day 14 were much higher than those in healthy mice. Anti-inflammatory factors in the butyrate treatment group were significantly increased compared with those in the untreated group (Figure 4F). Simultaneously, the inflammatory factors in the butyrate treatment group were significantly decreased compared with those in the untreated group (Figure 4G). The same situation was observed in the *R. intestinalis* treatment group. These results indicate that *R. intestinalis* may exert an anti-inflammatory effect through butyrate.

The H&E-stained sections of mouse colon derived from DSS + butyrate and DSS + *R. I* demonstrated that the colonic mucosa of mice was repaired by butyrate, and the anatomical structure curves recovered to a healthy status after treatment (Figure 4H). Further analysis showed that butyrate and *R. intestinalis* effectively reduced the DSS-induced inflammation score (Figure 4I).

### 3.5. Butyrate, a Metabolite of R. intestinalis, Promoted TLR5 Expression at the Transcriptional Level

Accumulating evidence has proven that gene transcription can be regulated by the butyrate produced by *R. intestinalis* and other bacteria [30,31]. The expression of TLR5 was restored in the UC mouse model treated with *R. intestinalis*. Therefore, we next investigated whether the increased expression of TLR5 in the UC mouse model was mediated by butyrate. HT29 cells were stimulated with increasing butyrate concentrations and treatment durations. The results showed that the expression of TLR5 was regulated by butyrate in a time- and concentration-dependent manner (Figure 5A,B). Twenty-four hours after treatment with 4 mM butyrate, the cells achieved the optimal and highest expression of TLR5, which was confirmed by RT–qPCR and immunofluorescence staining (Figure 5C). Transcriptional and translation inhibitors (ActD and CHX, respectively) were applied to explore the regulatory mechanism of butyrate on TLR5 expression. As a result, both the mRNA (Figure 5D) and protein (Figure 5E,F) levels of TLR5 in cells with and without butyrate were completely inhibited by the transcription inhibitor actinomycin. However, the translation inhibitor CHX did not decrease the protein level of TLR5 or the mRNA level. These results indicate that the regulation of TLR5 by butyrate occurs at the transcriptional level rather than the translational level.

### 3.6. Butyrate Promoted Sp3 Binding to the Promoter of TLR5

Sp3 was reduced by RNA silencing, which verified the role of Sp3 in the expression of TLR5 induced by butyrate. The silencing effect on Sp3 was validated by RT–qPCR (Figure 6A,B) and a Western blot (Figure 6C). RT–qPCR and Western blotting were performed to quantify the levels of TLR5 in Sp3-disrupted cells with and without butyrate. These results showed that the expression of the TLR5 gene or protein did not change regardless of whether butyrate was added to the Sp3-knockdown cells (Figure 6D,E). Proteins were extracted from HT29 cells treated with butyrate to evaluate the level of Sp3. The results showed that at the protein level, TLR5 was improved significantly in HT29 cells treated with butyrate compared with untreated HT29 cells or Sp3-knockdown cells. The ChIP-sequencing assay was also applied to further verify the Sp3 binding region within the TLR5 promoter, and the possible sequences were confirmed by ChIP–qPCR (Figure 6F). These results indicate that the transcription factor Sp3 was upregulated by butyrate produced by *R. intestinalis*, and the expression of TLR5 was significantly increased with enhanced promoter binding to Sp3.

## 4. Discussion

Recent studies have demonstrated that disturbance of the intestinal microbiota is one of the core factors underlying UC intestinal inflammation. Research has shown that many anaerobic symbiotic bacteria produce SCFAs while helping to relieve IBD symptoms and maintain the balance of intestinal microbiota [32,33]. The present study proposes that the *R. intestinalis* metabolite butyrate inhibits colon inflammation. To clarify the anti-inflammatory mechanism of butyrate in ulcerative colitis, we used HT29 cells for in vitro experiments. The results showed that butyrate can initiate TLR5 transcription through Sp3, upregulate TLR5 expression, and inhibit the expression and release of inflammatory factors (IL-6, IFN-γ, and TNF-α).

SCFAs are the main metabolites of the intestinal microbiota and a crucial link between the host and intestinal microbiota. Butyrate is a representative SCFA that exhibits anti-inflammatory ability by inhibiting the production of chemokines and stimulating the production of adhesion molecules. Previous studies have shown that a decrease in butyrate-producing bacteria in the intestine is related to metabolic and inflammatory diseases. In some cases, SCFAs have been reported to promote the emergence of inflammation [34,35]. Butyrate, an HDAC inhibitor, has been shown to promote or suppress gene expression [26,36]. Inactivated *Akkermansia muciniphila* treatment was reported to promote energy excretion in mouse feces, possibly involving a decrease in carbohydrate absorption and an increase in intestinal epithelial cell turnover. *R. intestinalis* and butyrate may serve as targets for inhibiting inflammation; probiotics with postbiotic potential have anti-inflammatory effects on the host. TLR5 is found on the basolateral side epithelial cells in the intestine. TLR5 is a conserved host pattern recognition receptor that can activate the innate immune response and recognize microbial-associated molecular patterns (MAMPs) to prevent microbial infection, inflammation, tissue damage, and radiation to the colon [37,38].

The expression of TLR5 was reduced in colitis mice, whereas the expression was increased after treatment with *R. intestinalis*. SCFAs are the main metabolites of *R. intestinalis* and provide a vital link between the intestinal microbiota and the host. Accumulating evidence suggests that the decrease in butyrate-producing bacteria in the gut is associated with metabolic and inflammatory diseases. SCFAs play multiple roles in inflammation and the immune response in various cells [39]. Butyrate, a representative SCFA, displays anti-inflammatory ability by inhibiting the production of chemokines and the expression of adhesion molecules induced by stimulation. In some cases, SCFAs have also been reported to promote the development of inflammation [34,35]. Butyrate, as an HDAC inhibitor, has been proven to both promote and repress gene expression [26,36]. In this research, symptoms were alleviated in DSS-induced UC mice through *R. intestinalis* treatment, which is a butyrate-producing strain, and TLR5 gene expression was significantly upregulated compared with that in untreated mice.

To determine the role of butyrate in IEC resistance to UC, the expression of TLR5 was quantified in HT29 cells supplemented with various butyrate concentrations. As a result, TLR5 gene and protein expression levels in HT29 cells were significantly upregulated by 4 mM butyrate. In addition, this upregulation was inhibited by transcriptional inhibitors, so it can be speculated that butyrate regulates TLR5 expression at the transcriptional level. TLRs are a type of conserved host pattern recognition receptors that recognize MAMPs from the microbiota and activate the innate immune response [37]. Activation of TLR5 by flagellin results in the rapid activation of genes that encode products designed to protect the host from invading microbes. Although TLR5 on the basolateral surface of intestinal epithelial cells is activated by flagellin, which is the main structural element of bacterial flagella, immune cells are upregulated and trigger most TLRs [40]. TLR5 expression in the intestinal epithelium protects against microbial infections, inflammation, tissue injury, radiation colitis, proapoptotic stimuli, and colon cancers [38]. Moreover, TLR5 is essential for maintaining gastrointestinal health, as suggested by the inability to manage commensal microbiota and the development of spontaneous colitis with the loss of IEC-specific TLR5 and TLR5-mediated gene expression in mice. In brief, the downregulation of TLR5 in IECs in the presence of disordered intestinal microbiota is a major cause of colitis, but *R. intestinalis* can induce an immune response by upregulating TLR5 expression through the production of butyrate.

The expressions of genes with butyrate-responsive elements in their promoters can often be mediated by butyrate through Sp1/Sp3 binding sites (e.g., the p21 gene). It has been proven that butyrate-treated cells increase Sp1 and Sp3 occupation at the ATP2A3 promoter and increase ATP2A3 mRNA expression in gastric and breast cancer cell lines [41]. We examined the expression of TLR5, Sp1, and Sp3 by transcriptome analysis. The results showed that TLR5 and Sp3 expression levels were higher in the control and DSS + *R* groups than in the DSS group. TLR5 was also found to be positively related to Sp3. The transcriptional regulation of TLR5 by Sp3 was verified by siRNA interference in vitro. The results showed that butyrate enhanced the expression of TLR5 by mediating the binding of Sp3 to the TLR5 promoter. ChIP-seq proved the binding of Sp3 to the TLR5 promoter. Butyrate can upregulate TLR5 expression by regulating the binding of the Sp3 transcription factor to the TLR5 promoter.

It is proposed for the first time that intestinal *R. intestinalis* is essential for the rehabilitation and repair of adhesive membrane barriers in patients with ulcerative colitis. Flagellin enhances the immune response of colonic epithelial cells, and this process is due to the induction of TLR5 expression by the butyrate produced by *R. intestinalis*. Our study provides a new reference for the mechanism by which intestinal *R. intestinalis* relieves ulcerative colitis. Sp3 of the Sp transcription factor family is involved in the butyrate-induced transcriptional activation of TLR5. According to reports, the targeted immunity of bacterial flagellin can improve the composition and functionality of gut microbiota, facilitating animal digestion and nutrient utilization. Flagellin can stimulate mucosal cells to produce chemokines, and TLR5-mediated signal transduction can promote cytokine production and cell recruitment and activation. Furthermore, flagellin stimulates mucosal cells to produce a variety of antimicrobial peptides in the mucosal lumen, including mucin and β-defensin, which aid in host immunological defense. Our data showed that butyrate enhanced the expression of TLR5 by mediating the binding of Sp3 to the TLR5 promoter. However, whether the effects of butyric acid and flagellin are consistent needs further research. In addition, this study only included six experimental animals during the in vivo assays, and a larger experiment on mice and other animal models of UC should be conducted to further identify the role of butyrate in regulating Sp3 expression and its anti-inflammatory effects. Additionally, we will further study the enzymes modified by Sp3 after butyrate treatment and hope to reveal that butyrate mediates the Sp3 regulatory mechanism. The transcriptional activation of TLR5 in cells inhibits the molecular mechanism of UC.

## 5. Conclusions

Overall, the present study showed that *R. intestinalis* plays a key role in alleviating the symptoms of UC. The flagellin of *R. intestinalis* can induce the immune response mediated by TLR5 and promote the release of anti-inflammatory factors from IECs. Additionally, butyrate produced by *R. intestinalis* can further increase the expression of the TLR5 gene and inhibit colitis. Therefore, these findings have significant implications for understanding the etiological mechanism and suggest a new treatment strategy for UC.

## Figures and Tables

**Figure 1 nutrients-14-03041-f001:**
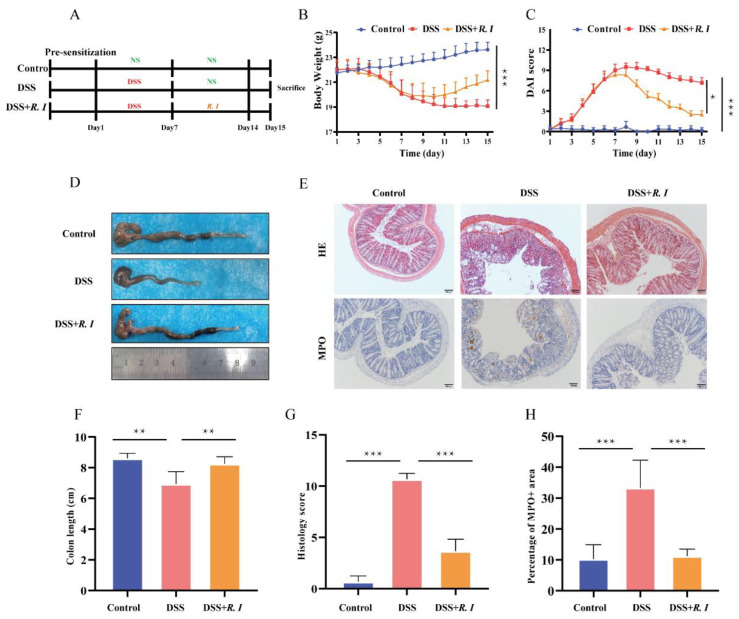
Mice with DSS-induced intestinal damage and clinical symptoms treated with *R. intestinalis*. (**A**) Schematic of the effect of *R. intestinalis* on DSS-induced colitis in mice. (**B**) The disease activity index (DAI) scores of the control, DSS, and DSS + *R. I* treated mice. (**C**) The body weights of mice in the three groups are plotted and presented as the mean. (**D**) Illustration of the colons of the mice in each of the three groups. (**E**) H&E staining for the three groups. Representative images of MPO immunohistochemical staining of colon sections from the three groups. (**F**) On Day 15, the colon lengths of the three groups were measured. (**G**) The histology score of H&E staining. (**H**) The percentage of MPO + area. The statistics are presented as the mean ± SD of three replicates; * *p* < 0.05, ** *p* < 0.01, and *** *p* < 0.001.

**Figure 2 nutrients-14-03041-f002:**
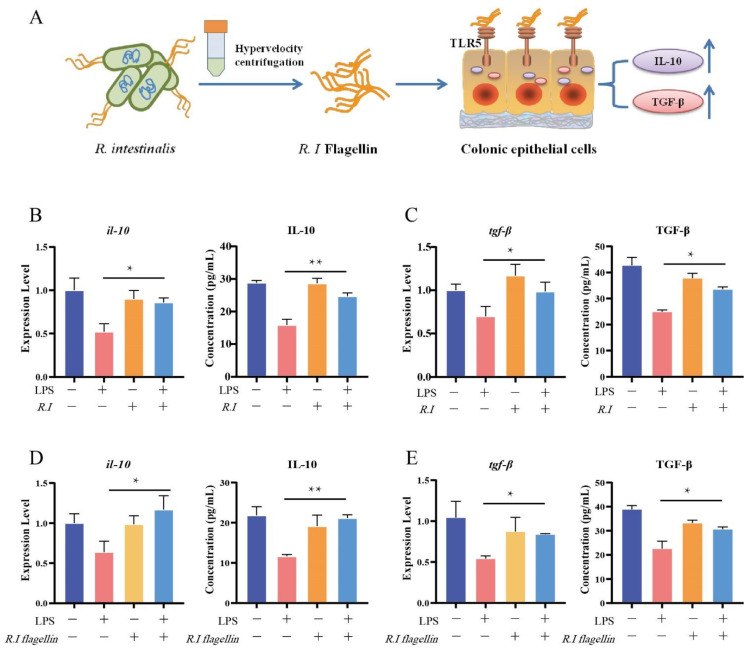
LPS-induced inflammation in HT29 cells ameliorated by *R. intestinalis* and flagellin of *R. intestinalis*. (**A**) Colonic epithelial cells with the TLR5 receptor recognize *R. intestinalis* flagellin and secrete the anti-inflammatory factors IL-10 and TGF-β. RT–qPCR and ELISA analysis of IL-10 (**B**) and TGF-β (**C**) mRNA and protein levels in the control, LPS, *R. I*, and LPS + *R. I* treated HT29 cells. RT–qPCR and ELISA analysis of IL-10 (**D**) and TGF-β (**E**) mRNA and protein levels in control, LPS, flagellin, and LPS + *R. I* flagellin treated HT29 cells. The results are presented as the mean ± SD of three replicates; * *p* < 0.05 and ** *p* < 0.01.

**Figure 3 nutrients-14-03041-f003:**
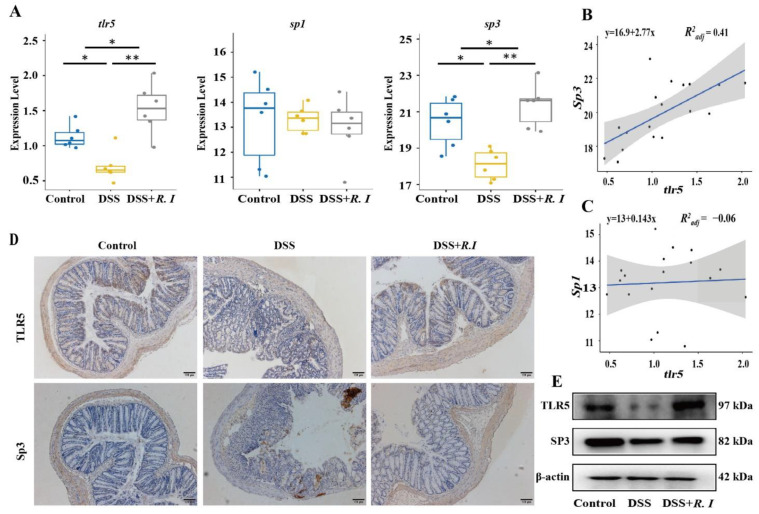
Expression levels of the flagellin recognition receptor TLR5 and its potential transcription factor in the control, DSS and DSS + *R. I* groups treated mice. (**A**) Expression levels of TLR5, Sp1, and Sp3 in the three groups as determined by RT–qPCR analysis. (**B**) Correlation analysis between TLR5 and Sp3 in the three groups. (**C**) Correlation analysis between TLR5 and Sp1 in the three groups. (**D**) Representative images of TLR5 and Sp3 immunohistochemical staining in colon sections from the three groups. (**E**) Representative Western blot analysis showing TLR5 and Sp3 in colon sections from the three groups. Control vs. DSS, Control vs. DSS + R. I, and DSS vs. DSS + R. I; * *p* < 0.05 and ** *p* < 0.01.

**Figure 4 nutrients-14-03041-f004:**
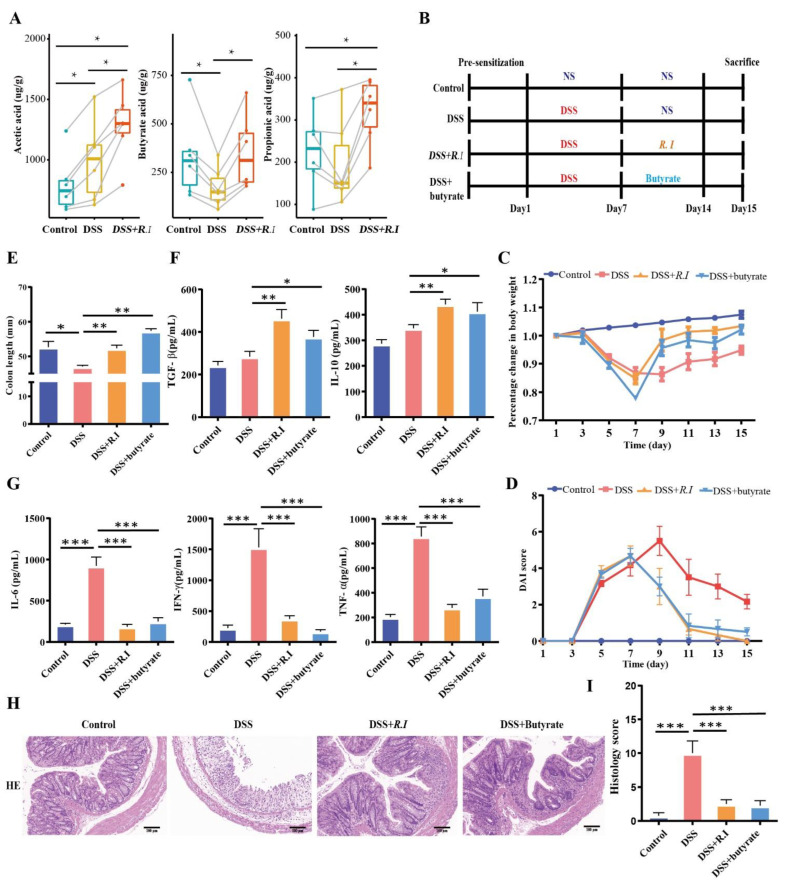
*R. intestinalis* inhibits the inflammatory progression of UC through its metabolite butyrate. (**A**) SCFA profiling in the control, DSS, DSS + *R. I*, and DSS + butyrate groups. (**B**) Schematic testing of the effect of *R. intestinalis* and butyrate on colitis mice generated by DSS. (**C**) The percent change in body weights of mice in the three groups is plotted and presented. (**D**) The DAI scores of the control, DSS, DSS + *R. I*, and DSS + butyrate groups treated mice. (**E**) The colon length of the three groups on Day 15. (**F**,**G**) Multifactor analysis of TGF-β, IL-10, IL-6, IFN-γ, and TNF-α protein levels in the control, DSS, DSS + *R. I*, and DSS + butyrate groups. (**H**) Representative images of H&E staining among the three groups. (**I**) The histology score of HE staining. These data represent the mean ± SD of 3 replicates; * *p* < 0.05, ** *p* < 0.01, and *** *p* < 0.001.

**Figure 5 nutrients-14-03041-f005:**
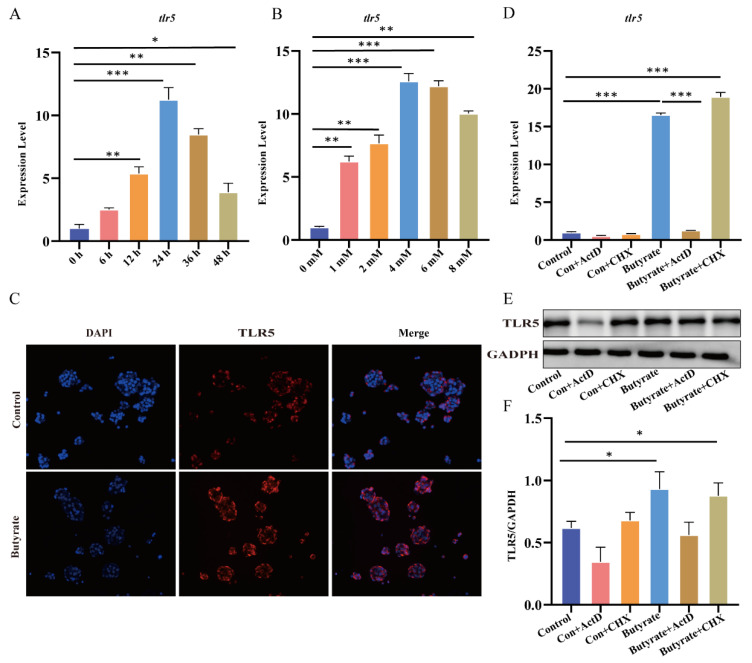
Butyrate increases TLR5 transcriptional expression in HT29 cells. RT–qPCR shows TLR5 mRNA expression in HT29 cells treated with a physiological concentration of butyrate (1 to 8 mM) for 24 h (**A**) or with 4 mM butyrate for a specific duration (**B**). (**C**) Immunofluorescence images of TLR5 in the control and butyrate groups; the butyrate treatment conditions were 4 mM for 24 h. (**D**) The change in TLR5 mRNA expression from adding transcription inhibitors or translation inhibitors in the control and butyrate groups. (**E**) Representative Western blot analysis showing TLR5 expression after the addition of transcription inhibitors or translation inhibitors to the control and butyrate groups. (**F**) Quantitative analysis of Western blot analysis. * *p* < 0.05, ** *p* < 0.01, and *** *p* < 0.001.

**Figure 6 nutrients-14-03041-f006:**
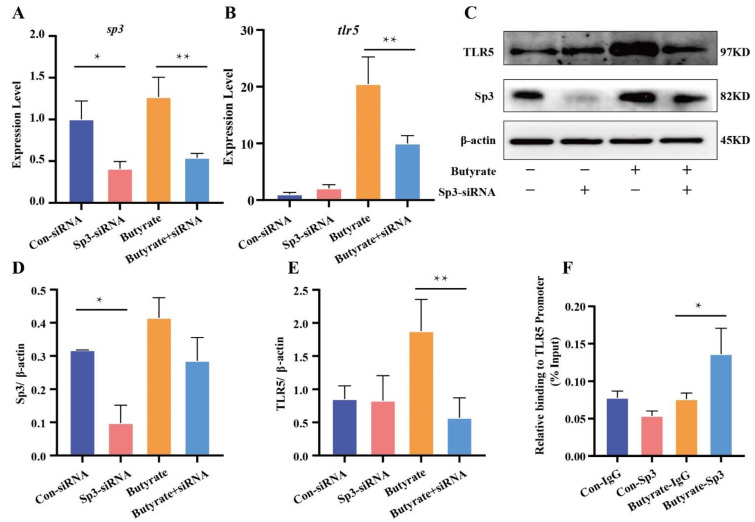
Butyrate regulates TLR5 expression through the transcription factor Sp3. (**A**) Control siRNA and Sp3-specific siRNA with and without butyrate were transfected into HT29 cells. After 48 h, Sp3 mRNA expression and TLR5 mRNA expression (**B**) were detected by RT–qPCR. (**C**) Western blot analysis and quantification of TLR5 and Sp3 protein expression after transfection with control siRNA and Sp3-specific siRNA in the control and butyrate groups. (**D**,**E**) A representative image of Western blot analysis. (**F**) ChIP-sequencing analysis of Sp3 in HT29 cells treated with butyrate. ChIP assays with H3 and Sp3 antibodies in butyrate-treated or untreated HT29 cells. qPCR showed H3 and Sp3 binding to the TLR5 promoter or RPL30 promoter. * *p* < 0.05 and ** *p* < 0.01.

**Table 1 nutrients-14-03041-t001:** Primer sequences used for RT–qPCR.

Gene	Sequences (5′-3′)
Human gapdh	TGTGGGCATCAATGGATTTGG
	ACACCATGTATTCCGGGTCAAT
Human il-10	GAACCAAGACCCAGACATCAAG
	GCATTCTTCACCTGCTCCAC
Human tgf β	CTAATGGTGGAAACCCACAACG
	CAGCAACAATTCCTGGCGAT
Human tlr5	CAACTTGCCTGGGAAACTGA
	GATGGCTAAATACTCCTGGTGG
Human sp3	CTGTAAAGAAGGTGGTGGAAGAG
	AGAATGCCAACGCAGATGAG
Mouse gapdh	TGAAGCAGGCATCTGAGGG
	CGAAGGTGGAAGAGTGGGAG
Mouse sp3	GGGAAGGACATACTGCCCAC
	TGCTTTTGTGTTGGAACTCTTGT
Mouse sp1	CGGTCCGGGTTCGCTTG
	ACCCCCATTATTGCCACCAA
Mouse tlr5	GGCAAGGTTCAGCATCTTCAA
	GGCAAGGTTCAGCATCTT

## Data Availability

The data presented in this study are available on request from the corresponding author.

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
