# Peer review of "Roseburia intestinalis and Its Metabolite Butyrate Inhibit Colitis and Upregulate TLR5 through the SP3 Signaling Pathway"

_nutrients, 2022, doi:10.3390/nu14153041_

Round 1

Reviewer 1 Report

The study focused on The metabolite butyrate of Roseburia intestinalis on UC. It was confirmed that The metabolite butyrate of Roseburia intestinalis up-regulates TLR5 expression and inhibits inflammatory progression through the SP3 signaling pathway. This work provides a new insight for understanding the mechanism of how R. intestinalis act in improving UC and provides a potential target for the prevention and treatment of UC. The work is very interesting. However, there are many mistake.

1. Abstract: “FMT (Fecal microbiota transplantation) can relieve intestinal mucosal inflammation in UC” The article mainly studies the anti-inflammatory mechanism of single bacterial colonization, and does not involve fecal bacterial transplantation, and the research background is confusing.

Abstract: “Overall, we confirmed that R. intestinalis and TLR5 played important roles in reducing inflammation by promoting TLR5 expression.”it is confused about “TLR5 ”

2 The position on line 111 treated with 200 μL of 1×109 CFU R. intestinalis”

109 Writing error

3 The position on line 119    Disease activity index 

References to disease activity indexes need to be listed

4 The position on line 140-147 

Missing gene primer sequences

 5 The position on line 152-153,158

Antibody dilution factor is unclear

6 The position on line 170

The concentration range of ELISA kit was unknown

7 The position on line 170 “buffer (pH 7.3) and centrifuged at 4C at 16,000 x g for 15 min.

Writing error

8 Materials and Methods section

 Many kits are missing the item number label.

9 Results section heading 3.1, 3.2, 3.4, 3.6 

Grammatical tense error

10 The position on line 247 (1×108 CFU/1×106 HT29 cells)

106 Writing error

11 The study found that Flagellin from R. intestinalis displays anti-inflammatory effects in HT29 cells and R. intestinalis inhibited the inflammatory progression of UC through its metabolite butyrate. So which one has the main anti-inflammatory effect? Do they regulate each other?

12 The position on line 340-341

“However, the translation inhibitor CHX only decreased the protein level of TLR5, but not the mRNA level.”

Results Figure 5E, F did not show that CHX reduced the expression of TLR5 protein level.

13 Discussion section

The result showed that butyrate enhanced the expression of TLR5 by mediating the binding of Sp3 with the TLR5 promoter, how does the anti-inflammatory effect of flagellin activate TLG-5-mediated, transcriptional level or protein level? Is it consistent with the effect of butyric acid?

14 the manuscript needs to be carefully revised with word spelling and grammar.

Author Response

Dear Reviewer:

Thanks for your comments on our manuscript entitled “The metabolite butyrate of Roseburia intestinalis up-regulates TLR5 expression and inhibits inflammatory progression through the SP3 signaling pathway” (ID: 1799564). Those comments are all valuable and very helpful for revising and improving our paper, as well as the important guiding significance to our research. We have studied the comments carefully and have made the correction which we hope meet with approval. The main corrections in the paper and the responses to the reviewers’ comments are as follows:

1. Abstract: “FMT (Fecal microbiota transplantation) can relieve intestinal mucosal inflammation in UC” The article mainly studies the anti-inflammatory mechanism of single bacterial colonization, and does not involve fecal bacterial transplantation, and the research background is confusing.

Abstract: “Overall, we confirmed that R. intestinalis and TLR5 played important roles in reducing inflammation by promoting TLR5 expression.” it is confused about “TLR5”.

Thanks for the comment. To make the paper more logical, we removed the FMT content and emphasized the background introduction of Roseburia intestinalis in the abstract. The Abstract has been revised.

2. The position on line 111 “treated with 200 μL of 1×109 CFU R. intestinalis”

109 Writing error

Thanks for this comment. We have corrected it according to the Reviewer’s comments. This position is on line 110.“200 μL of 1×109 CFU R. intestinalis.”

3. The position on line 119    Disease activity index.

References to disease activity indexes need to be listed.

Thanks for this comment. We are very sorry for our negligence with the reference [29]. The manuscript has been revised. This position is on line 117-122.

4. The position on line 140-147

Missing gene primer sequences

Thanks for this comment. The manuscript has been revised. The primers are seen in Table 1. This position is on line 148.

5. The position on line 152-153,158

Antibody dilution factor is unclear

Thanks for this comment. The manuscript has been revised. This position is on line 150-157. “Blocked membranes were incubated with specific primary antibodies (GAPDH, 1:2000, ab181602, Abcam, USA; β-actin, 1:2000, 3700S, CST, USA; TLR5,1:1000, ab13876, Abcam, USA; Sp3, 1:1000, sc-28305, Santa Cruz, USA) overnight at 4℃. Corresponding HRP-labeled IgG (IgG; 1:1000, ab172730 Abcam, USA) was applied to detect the primary antibody on PVDF membranes. The sections were developed with a detection kit (Thermo, USA).”

6. The position on line 170

The concentration range of ELISA kit was unknown

Thanks for this comment. The manuscript has been revised. The IL-10 (Cat. Number: PI528) and TGF-β (Cat. Number: PT880) proteins were detected by ELISA kits according to the procedures in the experimental guidelines (Beyotime, China). The minimum detection amount was 8.8pg/ml and 1.8pg/ml, respectively. This position is on lines 173-176.

7. The position on line 170 “buffer (pH 7.3) and centrifuged at 4C at 16,000 x g for 15 min.”

Writing error

Thanks for this comment. We have corrected it according to the Reviewer’s comments. This position is on line 201. “buffer (pH 7.3) and centrifuged at 4℃ at 16,000 x g for 15 min”

8. Materials and Methods section

Many kits are missing the item number label.

Thanks for this comment. We have added missing reagent numbers and manufacturers in the manuscript.

9. Results section heading 3.1, 3.2, 3.4, 3.6

Grammatical tense error

Thanks for this comment. The manuscript has been revised.

3.1. DSS-induced intestinal inflammation relieved by R. intestinalis

3.2. Flagellin from R. intestinalis displayed anti-inflammatory effects in HT29 cells

3.4. R. intestinalis inhibited the inflammatory progression of UC through its metabolite butyrate

3.6. Butyrate promoted Sp3 binding to the promoter of TLR5

10. The position on line 247 (1×108 CFU/1×106 HT29 cells)

106 Writing error

Thanks for this comment. The manuscript has been revised. “1×108 CFU/1×106 HT29 cells”. This position is on line 1087.

11. The study found that Flagellin from R. intestinalis displays anti-inflammatory effects in HT29 cells and R. intestinalis inhibited the inflammatory progression of UC through its metabolite butyrate. So, which one has the main anti-inflammatory effect? Do they regulate each other?

Thanks for this comment. The effect of flagellin in anti-inflammatory regulation is through TLR5 receptor activation, this is not found by our work, as far as we know, the R. intestinalis strain regulate anti-inflammatory cytokines in UC mice and LPS-induced cell model, and the effect of R. intestinalis in regulating anti-inflammatory cytokines is stronger than single application of flagellin or butyrate. However, if there is any regulating effect between flagellin and butyrate is still unknown, further study will be performed in our future work to reveal if there is any connection between the anti-inflammatory effects of flagellin and butyrate.

12. The position on line 340-341

“However, the translation inhibitor CHX only decreased the protein level of TLR5, but not the mRNA level.”

Results Figure 5E, F did not show that CHX reduced the expression of TLR5 protein level.

Thanks for this comment from the reviewer, the manuscript was supposed to illustrate that transcriptional inhibitor has only down-regulated the mRNA and protein level of TLR5, while the translational inhibitor had no influences in TLR5 expression. The sentence in the former manuscript was a mistake, and has now been revised. This position is on lines 1200-1202.

13. Discussion section

The result showed that butyrate enhanced the expression of TLR5 by mediating the binding of Sp3 with the TLR5 promoter, how does the anti-inflammatory effect of flagellin activate TLR-5-mediated, transcriptional level or protein level? Is it consistent with the effect of butyric acid?

Thanks for this comment. According to reports, the targeted immunity of bacterial flagellin can improve the composition and functionality of gut microbiota, facilitating animal digestion and nutrition utilization. TLR5 can distinguish between probiotic flagellin and pathogenic bacteria flagellin to achieve the goals of eradicating pathogenic bacteria and accepting probiotics. As a result of detecting flagellin, it can activate an immunological response to pathogenic bacteria and immune tolerance to probiotics. Flagellin can stimulate mucosal cells to produce chemokines, and TLR5-mediated signal transduction can promote cytokine production as well as cell recruitment and activation. Furthermore, flagellin stimulates mucosal cells to produce a variety of antimicrobial peptides in the mucosal lumen, including mucin and β-defensin, which aid in host immunological defense. Our data showed that butyrate enhanced the expression of TLR5 by mediating the binding of Sp3 with the TLR5 promoter. However, whether the effect of butyric acid and flagellin is consistent needs further research reports. We added this part in Discussion section. This position is on the lines 1534-1543.

14. The manuscript needs to be carefully revised with word spelling and grammar.

Thanks for this comment. The manuscript has been revised.

We tried our best to improve the manuscript and made some changes in the manuscript. These changes will not influence the content and framework of the paper. Any changes we make to the manuscript can be viewed in the "Track changes". We appreciate for your warm work earnestly and hope that the correction will meet with approval. Once again, thank you very much for your comments and suggestions.

If you have any questions or issues, please do not hesitate to email me back.

Yours sincerely,

Corresponding author: Yanling Wei, M.D., Ph.D.

[email protected]

Reviewer 2 Report

This is an interesting work, which discuss the role of R. intestinalis in defending UC, and discuss the potential mechanism. Below is my comments:

Changed flora to microbiota in the whole paper.

Italic Bacillus subtilis, Bifidobacterium longum in the whole paper, please italic bacteria names at genus and species level in the whole paper.

Transplantation of R. intestinalis is not righy, the single strain is not Transplantation

Only 6 mice were used in each group, listed this as a limitation in the paper

Authors indicated that butyrate play the key role as the main metabolites of the R. intestinalis, but other intestinal strain also produce butyrate, so how to exclude their impact.

How to determine the use amounts of R. intestinalis and butyrate?

CO2 check this

Provide the full name of the abbreviation when they first appear in the paper.

4C check this, please use the oC in the whole paper

Authors indicated that R. intestinalis inhibit the inflammatory progression of UC through its metabolite butyrate, and Flagellin from R. intestinalis displays anti-inflammatory effects in HT29 cells. So which one play the key role in defending the inflammatory.

Author Response

Dear Reviewer:

Thanks for your comments on our manuscript entitled “The metabolite butyrate of Roseburia intestinalis up-regulates TLR5 expression and inhibits inflammatory progression through the SP3 signaling pathway” (ID: 1799564). Those comments are all valuable and very helpful for revising and improving our paper, as well as the important guiding significance to our research. We have studied the comments carefully and have made the correction which we hope meet with approval. The main corrections in the paper and the responses to the reviewers’ comments are as follows:

1. Changed flora to microbiota in the whole paper.

Thanks for this comment. We are very sorry for our incorrect writing. The manuscript has been revised.

2. Italic Bacillus subtilis, Bifidobacterium longum in the whole paper, please italic bacteria names at genus and species level in the whole paper.

Thanks for this comment. The manuscript has been revised. This position is on line 43 and 159.

3. Transplantation of R. intestinalis is not righty, the single strain is not Transplantation

We appreciated this comment from reviewer, since R. intestinalis is a microbial strain, it may not be transplanted but administrated, the manuscript has been revised, and the Transplantation has been revised to Administration.

4. Only 6 mice were used in each group, listed this as a limitation in the paper

Thanks for this comment, a larger scale of animal experiments would be carried out in our future work to further identify the role of butyrate in anti-inflammatory effect against UC, and this limitation has been added in the revised manuscript. This position is on lines 1543-1546.

5. Authors indicated that butyrate play the key role as the main metabolites of the R. intestinalis, but other intestinal strain also produce butyrate, so how to exclude their impact.

Thanks for this comment. First of all, this study was based on the research group's previous research finding that R. intestinalis can treat UC. We compared the changes of short-chain fatty acid metabolites in the stool of the DSS group and the R.I-treated group, and butyrate was significantly different between the two groups. Literature showed that the flagellin of R. intestinalis bacteria inhibits inflammation (Wu X, Pan S, Luo W, et al. Roseburia intestinalis‑derived flagellin ameliorates colitis by targeting miR‑223‑3p‑mediated activation of NLRP3 inflammasome and pyroptosis. Mol Med Rep. 2020;22(4):2695-2704. doi:10.3892/mmr.2020.11351; Quan Y, Song K, Zhang Y, et al. Roseburia intestinalis-derived flagellin is a negative regulator of intestinal inflammation. Biochem Biophys Res Commun. 2018;501(3):791-799. doi:10.1016/j.bbrc.2018.05.075). Therefore, we proposed that R. intestinalis treatment of UC may be through butyrate production to up-regulate the expression of TLR5 in epithelial cells, and TLR5 is activated by flagellin as a ligand to initiate downstream anti-inflammatory signals. Therefore, we did not give more consideration to the influence of other butyric acid-producing bacteria.

6. How to determine the use amounts of R. intestinalis and butyrate?

In our past study, DSS-induced UC model mice received 109 CFU/100 µl of R. intestinalis enterally by enema for 7 days, which effectively relieve inflammation (Xu, Fenghua et al. New pathway ameliorating ulcerative colitis: focus on Roseburia intestinalis and the gut-brain axis. Therapeutic advances in gastroenterology vol. 14 17562848211004469. 15 Apr. 2021, doi:10.1177/17562848211004469). In this experiment, we also used 109 CFU/100 µl R. intestinalis for treatment, and achieved the same therapeutic effect. We refer to the literature for the dosage of butyrate, in which the author used a therapeutic concentration of 100ug/g and effectively relieved enteritis. (Lee, Changhyun et al. Sodium butyrate inhibits the NF-kappa B signaling pathway and histone deacetylation, and attenuates experimental colitis in an IL-10 independent manner. International immunopharmacology vol. 51 (2017): 47-56. doi:10.1016/j.intimp.2017.07.023). We added the two references in the revised manuscript. This position is on lines 403 and 405.

7. CO2 check this

Thanks for this comment. The manuscript has been revised. The position on line 430.

8. Provide the full name of the abbreviation when they first appear in the paper.

Thanks for this comment. The manuscript has been revised.

9. 4C check this, please use the oC in the whole paper

Thanks for this comment. The manuscript has been revised.

10. Authors indicated that R. intestinalis inhibit the inflammatory progression of UC through its metabolite butyrate, and Flagellin from R. intestinalis displays anti-inflammatory effects in HT29 cells. So, which one play the key role in defending the inflammatory.

Several reports showed that the R. intestinalis strain can regulate anti-inflammatory response through TLR5 activation by its flagellin, in our work, we found that the strain can contribute to the repairment of adhesive membrane barriers through its metabolite butyrate in UC. However, our current work did not study which part of the strain played a more important role in the anti-inflammatory response, further study will be conduct to reveal the most essential component and mechanism of the R. intestinalis strain on alleviating UC.

We tried our best to improve the manuscript and made some changes in the manuscript. These changes will not influence the content and framework of the paper. Any changes we make to the manuscript can be viewed in the "Track changes". We appreciate for your warm work earnestly and hope that the correction will meet with approval. Once again, thank you very much for your comments and suggestions.

If you have any questions or issues, please do not hesitate to email me back.

Yours sincerely,

Corresponding author: Yanling Wei, M.D., Ph.D.

[email protected]
